# Using Multiplex Amplicon PCR Technology to Efficiently and Timely Generate Rift Valley Fever Virus Sequence Data for Genomic Surveillance

**DOI:** 10.3390/v15020477

**Published:** 2023-02-09

**Authors:** John Juma, Samson L. Konongoi, Isidore Nsengimana, Reuben Mwangi, James Akoko, Richard Nyamota, Collins Muli, Paul O. Dobi, Edward Kiritu, Shebbar Osiany, Amos A. Onwong’a, Rachael W. Gachogo, Rosemary Sang, Alan Christoffels, Kristina Roesel, Bernard Bett, Samuel O. Oyola

**Affiliations:** 1International Livestock Research Institute (ILRI), Nairobi P.O. Box 30709, Kenya; 2South African National Bioinformatics Institute (SANBI), University of the Western Cape, Cape Town 7535, South Africa; 3Centre for Virus Research, Kenya Medical Research Institute (KEMRI), Nairobi P.O. Box 54840, Kenya; 4Department of Microbiology, Parasitology and Biotechnology, Sokoine University of Agriculture, Morogoro 3019, Tanzania; 5Rwanda Inspectorate, Competition and Consumer Protection Authority, Kigali P.O. Box 375, Rwanda; 6Department of Veterinary Services, Ministry of Agriculture, Livestock and Fisheries, Nairobi P.O. Box 30028, Kenya; 7Department of Medical Laboratory Sciences, Jomo Kenyatta University of Agriculture and Technology (JKUAT), Nairobi 00200, Kenya; 8Division of Immunology, Department of Human Pathology, University of Cape Town, Cape Town 7925, South Africa

**Keywords:** amplicon, multiplex PCR, enrichment, culture, genome, Rift Valley fever

## Abstract

Rift Valley fever (RVF) is a febrile vector-borne disease endemic in Africa and continues to spread in new territories. It is a climate-sensitive disease mostly triggered by abnormal rainfall patterns. The disease is associated with high mortality and morbidity in both humans and livestock. RVF is caused by the Rift Valley fever virus (RVFV) of the genus *Phlebovirus* in the family *Phenuiviridae*. It is a tripartite RNA virus with three genomic segments: small (S), medium (M) and large (L). Pathogen genomic sequencing is becoming a routine procedure and a powerful tool for understanding the evolutionary dynamics of infectious organisms, including viruses. Inspired by the utility of amplicon-based sequencing demonstrated in severe acute respiratory syndrome coronavirus-2 (SARS-CoV-2) and Ebola, Zika and West Nile viruses, we report an RVFV sample preparation based on amplicon multiplex polymerase chain reaction (amPCR) for template enrichment and reduction of background host contamination. The technology can be implemented rapidly to characterize and genotype RVFV during outbreaks in a near-real-time manner. To achieve this, we designed 74 multiplex primer sets covering the entire RVFV genome to specifically amplify the nucleic acid of RVFV in clinical samples from an animal tissue. Using this approach, we demonstrate achieving complete RVFV genome coverage even from samples containing a relatively low viral load. We report the first primer scheme approach of generating multiplex primer sets for a tripartite virus which can be replicated for other segmented viruses.

## 1. Introduction

Rift Valley fever (RVF) is a zoonotic disease caused by an arthropod-borne RNA virus, namely the Rift Valley fever virus (RVFV) of the genus *Phlebovirus* in the family *Phenuiviridae* [1]. The virus was first discovered in the 1930s following an outbreak linked with high rates of abortion among pregnant sheep and acute deaths of newborn lambs on a farm near Lake Naivasha in the Rift Valley region of Kenya [2]. The disease is responsible for deaths in humans and animals, as witnessed in some of the largest outbreaks in Africa and the Arabian Peninsula [3]. RVF has huge economic implications in livestock production, thus negatively impacting livelihoods in sub-Saharan Africa [4,5,6,7]. RVF can be transmitted to humans through inoculation via a wound from contaminated surgical instruments, contact with infected material or inhalation of aerosolized particles while handling infected animals. In humans, the disease manifests itself as an acute febrile illness which can progress to severe disease characterized by hemorrhagic fever, encephalitis and ocular disease [7,8,9]. Among several diseases identified by the World Health Organization (WHO), RVF has been listed as a likely cause of a future epidemic in a new emergency plan developed after the Ebola epidemic [10]. Consequently, it has been prioritized for urgent research towards new diagnostic tests, vaccines and medicines [11].

A powerful tool which has become useful in studying emerging and re-emerging infectious diseases is genomic sequencing [12,13]. By sequencing pathogens, we can answer different diagnostic questions such as the genetic relationship of viruses and the detection of mutations in viral genomes, which potentially lead to increased virulence, resistance towards antivirals, vaccine failure or immune escape [14,15]. Genome sequencing of pathogens can be performed directly on clinical samples in an unbiased way [16] through a metagenomics approach or following enrichment methods that include cell culture [14,17]. In circumstances where the etiological agent of an outbreak is unknown, metagenomics is the option of choice. Metagenomics is therefore well suited when dealing with potentially lethal infections that fail the conventional diagnostic procedures such as immunoenzyme and immunofluorescence methods. However, sequencing an entire sample through metagenomics is less sensitive, generating insufficient pathogen reads due to the abundance of host contamination and thereby yielding fragmented and incomplete genome sequences [14,16,18]. Genome sequencing directly from clinical samples without isolation is very difficult for viruses such as RVFV whose viremia levels drops 8 days post-infection [8,19].

In trying to sequence Zika virus using a metagenomics approach, Quick and colleagues were unable to recover sufficient sequence reads of the virus even after depleting human ribosomal RNA (rRNA) due to the low levels of viremia (<1000 copies/µL of RNA) [20]. In many cases, measuring virus diversity with high accuracy using deep sequencing is a challenging task. Factors such as virus titer, sample preparation, sequencing errors and computational inferences can bias measures of genetic diversity [21,22,23]. Detection of pathogens through metagenomics is made more complex by specificity issues that arise from misclassification or contamination, nucleic acid stability, evolving bioinformatics workflows and high costs involved in data generation and analysis [18,24]. Therefore, a timely response in terms of sample collection and processing is very critical to enhance recovery of virus genomes, especially for RNA viruses.

To undertake real-time genomic surveillance, there is a need to rapidly sequence the viral material directly from clinical samples without cell culture enrichment, which is laborious and time-consuming. To generate complete viral genomes from clinical samples economically, targeted enrichment is needed [15,20]. Targeted enrichment can be achieved indirectly via host nucleic acid depletion or directly using oligonucleotide bait probes targeting the virus of interest [15]. Sequencing of complete genomic segments of RVFV has mostly relied on isolation by cell culture on VeroE6 (African green monkey) cell lines [25,26]. The manipulation of RVFV cultures is conducted in high containment biosafety laboratories and requires highly skilled personnel. Moreover, the cell culture process is often time-consuming (takes up to 14 days) and laborious [15]. Viral passaging can also introduce mutations that were not present in the original clinical sample, therefore leading to incorrect variant determination [27].

An alternative approach to cell culture is enrichment using the amplicon multiplex PCR (amPCR) method in a single assay. Amplification of viral genetic material is achieved using primers that are complementary to a known nucleotide sequence. This targeted sequencing approach has been successfully used in enriching viral genomes such as Ebola [17], Zika [20,28,29] and currently SARS-CoV-2 [30]. Heterogeneous and segmented RNA viruses may necessitate the use of multiple overlapping sets of primers to ensure the amplification of all genotypes and segments. Although amplicon sequencing can also generate incomplete coding sequences of viral genomes when the viral load in a sample is low, such sequences can still be used for genotyping and other evolutionary analyses. Amplicon multiplex PCR-based sequencing offers many advantages, including the following: (i) high specificity, as most sequence reads will be of pathogen origin and not the host, which significantly reduces the sequencing cost; (ii) high sensitivity, with good coverage even at low pathogen load; and (iii) simplicity in terms of the design and application of new sets of primers for novel sequences [15]. Therefore, amplicon enrichment methods are relatively cheap, readily available and a fast option in comparison to isolation by cell culture, which is time-consuming, expensive and laborious. To generate RVFV genomes in a cost-effective and timely manner, we have devised an amplicon primer scheme—the very first for a segmented genome—and show that high-quality genomes with high accuracy and good coverage are achievable. Given the advantages of amplicon multiplex PCR-based sequencing, we aimed to reconstruct RVFV genomes with higher coverage and depth, which are crucial for genotyping and evolutionary inference.

## 2. Materials and Methods

### 2.1. Samples Dataset

For the purpose of benchmarking the amplicon multiplex PCR-based sequencing approach, we subjected 5 livestock (bovine) samples to three different treatments, namely amplicon multiplex PCR enrichment, cell culture enrichment and non-enrichment (Direct), representing the different enrichment/non-enrichment processes undertaken, as indicated in Table 1.

In summary, samples were subjected to three categories of treatment, including amplicon multiplex PCR (*n* = 5), cell culture (*n* = 5) and non-enrichment (*n* = 5). In the sample set, 1 sample (RU1) was collected from an outbreak event in Rwanda, while the rest were all from Kenyan outbreaks. The year of sample collection ranged between 2018 and 2021. Appendix A provides detailed information on the samples used in the study. Clinical outbreak samples from Kenya were linked to a record that collated both the epidemiological and clinical data, including the date of sample collection, date of onset of symptoms, geographical location and demographic characteristics. The affected animals displayed symptoms such as lack of appetite, dyspnea, extreme fever, ecchymosis, conjunctiva, reddish vulva, diarrhea, jaundice, congestion of udder, nasal discharge and abortion.

### 2.2. IgM Antibody Capture ELISA

All the RVF-suspected livestock samples collected in Kenya in 2021 were first screened using enzyme-linked immunosorbent assay (ELISA) for the detection of anti-nucleoprotein IgM antibodies in sera. This was conducted using the ID Screen^®^ RVF IgM Capture MAC ELISA as per the manufacturer’s instructions (IDVet Innovative Diagnostic, Grabels, France) as previously described [31].

### 2.3. Virus Enrichment by Cell Culture

Virus culture enrichment was performed in a biosafety laboratory level 3 (BSL3) containment facility as previously described in our study [31].

### 2.4. Nucleic Acid Isolation

For the Direct and amPCRe samples, viral RNA was isolated from 140 µL IgM-positive samples using the NucleoSpin RNA Virus Mini kit for viral RNA (MACHEREY-NAGEL, Ref 740956, Dueren, Germany) according to the manufacturer’s instructions. Viral RNA was isolated from 140 µL of the filtered cell culture supernatant (CCE samples) using a QIAmp Viral RNA kit (QIAGEN, Hilden, Germany) according to the manufacturer’s instructions.

### 2.5. Diagnosis Using RT-qPCR

Reverse transcription quantitative real-time PCR (RT-qPCR) was performed on the RNA samples. The one-step assay comprised 2 µL (5–10 ng) of the RNA template in a reaction of 15 µL using a final concentration of 0.3 µM for primers and 0.1 µM for the probe in a PCR system (Applied Biosystems, Waltham, MA, USA). The reaction was carried out in a series of incubation steps as follows: 50 °C for 10 min, 95 °C for 2 min, 95 °C for 3 s and 60 °C for 30 s for 40 cycles. This assay uses a highly conserved domain located on the L-segment of the virus for RVFV detection (using 5′ Fam reporter dye and 3′ BHQ1 quencher dye) [32], as shown in Table 2.

### 2.6. Designing Multiplex Amplicon (Tiling) Primers

RVFV primer panels for multiplex PCR were designed using Primal Scheme [20] for each segment of the virus. Primal Scheme uses Primer3 [33] and applies a greedy algorithm to find primers for tiling amplicon generation using multiple reference genomes. For each segment, a set of 104 complete genomic RVFV sequences was obtained from the NCBI RefSeq database [34]. The whole genome sequences for each segment were separately concatenated into a single FASTA file. The ZH548 isolate/strain was used as the coordinate system for the primers. Multiple sequence alignment on the DNA sequences was performed using Clustal Omega [35]. Sequences with more than 5% divergence and 99–100% identity to other genomes were removed, resulting in a final set of 16, 15 and 19 for the L, M and S segments, respectively. Primal Scheme was executed to generate amplicon primers with an amplicon size of 400 bp. A total of 74 primers in two pools were generated (Figure 1). In silico prediction of the coverage for whole genome sequences was reported to be 97.31%, 98.84% and 96.80% for the L, M and S segments, respectively. The primers were synthesized by Macrogen Europe (Amsterdam, The Netherlands) and shipped to ILRI Nairobi, Kenya, followed by reconstitution in tris-ethylenediaminetetraacetic acid (EDTA) buffer. Detailed information providing the primer locations can be found in Appendix A.

### 2.7. cDNA Synthesis

Extracted RNA was converted to cDNA using the LunaScript RT Supermix Kit (New England Biolabs, Hitchin, UK) in a reaction of 10 µL according to the manufacturer’s instructions.

#### 2.7.1. Amplicon Multiplex PCR

The multiplex PCR was set up in two separate reactions, each having one of the two primer pools as indicated in Table 3. The reaction conditions were marked by low primer concentrations, long annealing times and high primer annealing temperatures. This enabled amplification of targets that cover whole genome in the two reactions. Briefly, lyophilized primers were reconstituted in tris-ethylenediaminetetraacetic acid (EDTA) (TE) buffer to attain concentrations of 100 µM. Two 1.5-milliliter Eppendorf tubes were labeled as pool 1 and pool 2. An equal volume (100 µM stocks) of each odd-numbered primer was added to pool 1, while all the even ones were added to pool 2. In total, there were 38 and 36 primers for pool 1 and pool 2, respectively. The primer pools were then diluted at a ratio of 1:10 with TE buffer to a working concentration of 10 µM. 

The reactions were carried out in a series of incubation steps as follows: initial denaturation at 98 °C for 30 s, then 35 cycles of denaturation at 95 °C for 15 s and annealing at 63 °C for 5 min, ending with a hold at 4 °C.

#### 2.7.2. Library Preparation Using NEBNext Ultra II DNA Library Prep Kit

Library preparation of the samples was performed using a NEBNext Ultra II DNA library prep kit (New England Biolabs, Ipswich, MA, USA). PCR products from the amplicon reaction in separate primer pools were cleaned up using AMPure XP purification beads (Beckman Coulter, High Wycombe, UK). The clean products were quantified using fluorimetry with the Qubit dsDNA Broad Sensitivity assay on the Qubit 2.0 instrument (Thermo Fisher Scientific, Waltham, MA, USA). The corresponding PCR products from each primer pool were pooled together and quantified using Qubit for library preparation. End repair and adaptor ligation was performed using the NEBNext Ultra II DNA Library Prep Kit for Illumina (New England Biolabs, Hitchin, UK). Adaptor-ligated clean-up was performed using AMPure XP purification beads followed by library PCR using the NEBNext Ultra II Q5 Master Mix, TruSeq Index Primers (i7) and Universal PCR Primers (i5) in a 50 µL reaction for 15 cycles. Equal volumes were pooled, after which a clean-up with AMPure XP purification beads was carried out, followed by quantification. Pooled libraries were denatured and diluted for loading on an Illumina sequencing instrument according to the manufacturer’s instructions.

For CCE and Direct samples, 12 µL of RNA (concentration of 5 ng–1 µg) was added to a probe hybridization buffer (2 µL) and NEBNext ribosomal RNA (rRNA) depletion buffer (3 µL). The mixture was thoroughly mixed and incubated on a thermocycler, with a denaturation step at 95 °C for 2 min followed by a ramp down step to 22 °C at a rate of 0.1 °C per second and a final hold at 22 °C. The samples were then subjected to digestion with DNase I. Purification of the samples was performed using NEBNext RNA sample purification beads (2.2× the sample volume). The purified RNA samples were prepared for first-strand synthesis by hybridizing to random primers (NEBNext) and incubated for 8 min at 94 °C. First-strand cDNA was synthesized using a NEBNext first-strand synthesis enzyme mix and placed on a thermocycler with the following conditions: 10 min at 25 °C, 15 min at 42 °C, 15 min at 70 °C and a final hold at 4 °C. Second-strand synthesis was immediately performed using the NEBNext second strand synthesis reaction buffer and enzyme mix. The samples were incubated for 1 h at 16 °C with the lid temperature off. Double-stranded cDNA was purified using NEBNext sample purification beads. The clean cDNA samples were end repaired and adaptor ligated using NEBNext adaptors diluted 200-fold with the dilution buffer. This was conducted since the concentrations of the RNA samples were low. The NEBNext USER enzyme was added to the ligation reaction mixture. Adaptor-ligated samples were cleaned using AMPure purification beads (0.9×). Enrichment of the adaptor-ligated cDNA through PCR was performed using NEBNext multiplex primers (forward and reverse primers combined). The thermocycler conditions were as follows: initial denaturation at 98 °C for 30 s for 1 cycle, denaturation at 98 °C for 10 s and annealing at 65 °C for 15 cycles, final extension at 65 °C for 5 min for 1 cycle and a hold at 4 °C. Clean-up of the PCR enriched libraries was performed using sample purification beads (0.9×). The libraries were then quantified using the Qubit dsDNA HS assay kit and normalized to obtain a concentration of 4nm. Pooled libraries were denatured and diluted for loading on an Illumina sequencing instrument following the manufacturer’s instructions.

#### 2.7.3. Generation of Consensus Sequences

To statistically compare the mapping metrics of the samples, we normalized each library as a relative proportion of the total library size. We obtained the reads for every sample in 20 million and randomly subsampled the resulting number from the raw reads. Consensus genomes for each barcoded RVFV sample were generated using the rvfv-amplicon-seq [36] nextflow pipeline that we developed. Raw demultiplexed reads in FASTQ format were assessed on quality using FastQC v0.11.9 [37]. Reads that had low quality scores and adaptor sequences were trimmed using fastp [38]. Alignment of the trimmed reads to the RVFV reference (ZH548) was conducted using bwa-mem [39]. Alignment statistics were computed with SAMtools [40], and only samples having a minimum mapping threshold of 200 reads were utilized in downstream analysis. Alignments containing amplicon primers were trimmed using iVar [41]. Using each segment reference (ZH548 strain) genome and the corresponding gene features file, variants were called on the pile-up generated from the filtered alignments. To generate consensus alleles, positions with ≥10× coverage and ≥20 base quality were considered. Regions that failed the above criteria as well as those in primer binding were masked with N characters. Genome-wide, amplicon mean and amplicon per base coverages were computed using BEDTools [42]. Variant effect prediction was performed using SnpEff [43] and filtered one line per variant using vcfEffOnePerLine.pl that comes bundled with SnpSift [44]. Consensus genomes for culture-enriched and non-enriched samples were generated using viclara [45]. In this analysis pipeline, we called the variants using bcftools while maintaining the minimum base quality at ≥25 and at ≥10× coverage. SNP concordance analysis was performed on samples that had high genome coverage (>90%) using the SnpSift concordance method [44].

### 2.8. Maximum Likelihood Estimation and Molecular Clock Phylogenetic Reconstruction

We retrieved 216 complete M segment genome sequences and concatenated them to the 11 RVFV genomes generated in this study. We selected the 11 genomes based on a threshold value of 80% on consensus genomes. The sequences were deduplicated to remove those with a similar base composition followed by multiple sequence alignment using MAFFT [46] after the removal of primer binding sites. After this filtering process, there were 196 sequences used in downstream analyses (Appendix A). We identified GTR (Generalized Time Reversible) [47] with gamma (Γ) substitution rate as the optimal evolutionary model using ModelTest-NG [48]. We applied this model to infer a maximum likelihood phylogenetic tree using IQTREE2 [49] with single branch support testing [50] and 1000 replications. We also assigned lineages to the complete M segment genome sequences using a tool we recently developed [31]. We assessed the temporal signal of the sequences by regressing the root-to-tip distance against sampling time in decimal years using TempEst [51]. Molecular clock analysis was performed using TreeTime [52] on the alignment using the collection years of the sequences as dates to generate a time-scaled phylogenetic tree.

## 3. Results

### 3.1. Sequencing and Consensus Genomes

The inverse relationship between genome coverage and RT-qPCR cycle threshold (Ct) values was quite distinct, as expected in all the samples. Generally, samples subjected to cell culture enrichment (CCE) yielded genomes with high coverage compared to amplicon multiplex PCR-enriched (amPCRe) and non-enriched (direct) samples (Figure 2A–C). For samples with Ct > 30, genome recovery from direct sequencing was low, ranging from 25% to 70%, and in one sample (08HAB), no genome was recovered in all three segments. In another sample (RU1), with a Ct value of 25.25, direct sequencing yielded only the S segment genome. Enrichment by cell culture significantly allowed for genome recovery in most samples that had Ct values of >25. However, in one sample (DK-B2), genome recovery failed due to fewer reads mapping to the virus genome. Amplicon multiplex PCR-enriched samples yielded genomes with >80% coverage. Interestingly, samples with Ct > 30 produced near-complete genomes compared to non-enriched samples. In addition, samples with Ct < 25 showed genome recovery of over 90%. In amPCRe samples, there was a general trend of fewer reads mapping at the extreme ends (both 3′ and 5′) of the genome segments. We also observed a distinctive drop in coverage between positions 840 and 900 in the small (S) segment in all treatments, with amPCRe samples showing a large drop in depth at these positions. The patterns of genome-wide depth distribution in amPCRe samples were almost identical across the segments, with distinctive drops in specific regions (Figure 2D–F).

Out of the five samples subjected to CCE, we recovered four (DVS-356, DVS-230, RU1 and 08HAB) RVFV genomes with a coverage of >99%. Of the five amPCRe samples, we obtained five RVFV genomes with a coverage ranging between 80% and 97% in all segments. From the non-enriched (Direct) samples, we recovered only two (DVS-356 and DVS-230) RVFV genomes with a coverage of >99%. Two samples (DK-B2 and RU1) yielded fragmented partial genomes for one or all segments. In total, we recovered 11 complete and near-complete (>80% genome coverage) RVFV genomes. All the genomes recovered were from samples with Ct values less than 35. To compare genome coverage metrics, we selected three samples (DVS-356, DVS-230 and DK-B2) that yielded RVFV genomes in amPCRe and Direct treatments. For CCE, we excluded DK-B2. Overall, the genome coverage in CCE samples was higher (~99%) compared to that in Direct and amPCRe samples. At the genome segment levels, there was no significant variation in genome coverage among CCE samples. The CCE samples showed genome coverages in L (*n* = 2, μ = 99.8%, σ = 0.14), M (*n* = 2, μ = 99.6%, σ = 0.00) and S (*n* = 2, μ = 99.6%, σ = 0.28). The observed mean genome coverages in Direct samples were 74.7% (*n* = 3, σ = 43.7), 77.2% (*n* = 3, σ = 39.0), and 90.9% (*n* = 3, σ = 15.7) for L, M and S segments, respectively. Among the amPCRe samples, the genome coverages reported were 94.3% (*n* = 3, σ = 1.99), 95.3% (*n* = 3, σ = 3.60) and 95.0% (*n* = 3, σ = 1.83) for L, M and S segments, respectively. Nonetheless, in all three categories, there were insignificant differences in the mean genome coverage across the three genomic segments, as reported by the *p*-values of 0.89, 1.0 and 0.79 for CCE, Direct and amPCRe, respectively (Figure 2G–I).

### 3.2. Performance of Amplicon Primers

We examined the performance of each amplicon primer by computing the mean amplicon coverage per segment using BEDTools [42]. We observed that the first and last amplicon primers for every genomic segment of the virus reported coverage dropout. The 5′ and 3′ ends of the genome segments were insufficiently covered due to the minimal number of reads mapping in these regions. Uneven amplicon primer coverage of genomes was also observed with distinct patterns especially in samples with very high Ct values (≥35). In both L and M segments, amplicon primers that showed significant coverage dropout were mostly observed in samples with high (≥35) Ct values. However, two primers in the S segment, S_2_RIGHT (spanning positions 709–739) and S_4_LEFT (spanning positions 957–979), reported a significant drop in coverage irrespective of the sample Ct values. Zooming into this region, we identified a homopolymer track which is marked by repetitive stretches of C’s. Overall, samples with low Ct values were well covered by the amplicon primers compared to samples with high Ct values (Figure 3A–C).

### 3.3. Amplification Accuracy Assessed by SNP Concordance

To assess the accuracy of amPCRe against CCE and Direct treatments, we compared the SNPs identified in each treated sample. To identify the mutations within the RVFV genome, we enumerated changes in the generated consensus genomes. SNP concordance analysis was limited to two samples (DVS-230 and DVS-356) that yielded high (>94%) coverage genomes in all three treatments. We observed a higher proportion of synonymous to non-synonymous mutations in the RVFV genome for all three treatments. From the consensus alleles, we identified shared synonymous mutations that were mainly occurring within the RVFV genes in CCE, amPCRe and Direct samples. We found 146, 89 and 39 intragenic SNPs in the L, M and S segments, respectively, that were in perfect concordance in all three categories of treatment.

We also observed 27 non-synonymous mutations classified as missense variants in all the genomic segments of RVFV. There was insignificant variation in the number of mutations observed in CCE, amPCRe and Direct samples. In the amPCRe samples, the number of synonymous and non-synonymous mutations reported was 337 and 37, respectively. In the CCE samples, 286 and 31 respective mutations were reported, while in the Direct samples, we enumerated 287 and 29 synonymous and non-synonymous mutations, respectively. Eleven non-synonymous mutations located in the RNA-dependent polymerase in the L segment were common in all three treatments. All four non-synonymous mutations (T715A, A1717G, A1911G and A1913G) observed in the CCE, amPCRe and Direct samples occurred within the glycoprotein Gn gene. We observed 11 non-synonymous mutations occurring in the S segment, of which 10 were found in the non-structural (NSs) gene and only 1 was observed in the nucleocapsid (NP) gene. On evaluating the suitability of amplicon-enriched samples for genetic studies, SNP concordance showed high concordance between Direct and CCE samples. Of all the SNPs called, we observed a concordance score of 100% in all three genomic segments of the virus genome. However, there was a slight reduction in the concordance scores between Direct and amplicon-enriched samples. We observed concordance scores of 98.42% (out of 191 SNPs), 99.31% (out of 147 SNPs) and 91.04% (out of 67 SNPs) in the L, M and S segments, respectively.

### 3.4. Similar Lineage Placement in CCE, amPCRe and Direct Genomes

To infer the evolutionary dynamics of RVFV, we performed a phylogenetic analysis to determine whether the consensus genomes generated by the three treatment approaches can be used for such inferences. We began by assigning lineages to the M segment genome sequences that were generated from the study together with those obtained from the public database NCBI GenBank. All the genome sequences (*n* = 11) generated from this study were confidently assigned to lineage C. In addition to the three (T715A, A1717G and A1913G) non-synonymous mutations found in the M segment, multiple sequence alignment indicated 43 synonymous mutations found in lineage C that were also present in the CCE, amPCRe and Direct samples (Figure 4). This observation indicates the suitability of the data generated by amPCRe for evolutionary dynamic studies and inference.

## 4. Discussion

Genomic surveillance has become an important tool for studying emerging and re-emerging infectious diseases [12,13]. Using genomic data, control measures can be developed, including diagnostic and vaccination reagents. Virus genome sequence data can be used to explore different areas such as the genetic relationship of viruses and mutations patterns that can potentially lead to increased virulence, resistance towards antivirals, vaccine failure or immune escape [14,15]. The challenge, however, is obtaining reasonably pure virus genetic material from a clinical sample that is low in host contamination to generate a whole viral genome sequence. Existing methods for enriching viral genetic material and reducing host contamination includes culturing techniques that are expensive, laborious and time-consuming. Here, we have developed an RVFV sample preparation method based on amplicon multiplex polymerase chain reaction enrichment (amPCRe) that allows for template enrichment and reduction of background host contamination. The technology can be implemented rapidly in a near-real-time manner to characterize and genotype RVFV during outbreaks.

Assessments of the levels of antibodies against the nucleoprotein for all the samples used in the study were positive. The results were further corroborated by RT-qPCR, which revealed the amount of target nucleic acid present in the samples, reported as Ct values. Ct values of less than 29 often indicate a strong positive reaction with reasonably high viral titers in the samples. A Ct value in the range of 30–37 shows positivity with low viral titers in the sample. Finally, Ct values > 37 indicate a weak positive reaction with very low viral titers in the sample. In all the samples used in this study, we encountered all three categories of Ct values. Applying the three treatment options (CCE, amPCRe and Direct), we took note of the recovery success rate of RVFV genomes with respect to the original sample Ct values. We noted that when the Ct values of samples are less than 25, enrichment of the samples with CCE or amPCRe does not result in any significant different in genome recovery of the virus. However, when the Ct values of samples are over 25, our proposed amPCRe method produces an improvement in genome recovery (Figure 2).

We sequenced and assembled complete RVFV genomes in 11 samples, of which 4 had been subjected to CCE, 5 to amPCRe and 2 were from direct clinical outbreak and archived samples. Samples that were enriched by CCE produced near-complete genomes and covered over 99% of the genome, while Direct samples displayed genome coverage values ranging from 75% to 90% (Figure 2). Amplicon-enriched (amPCRe) samples yielded a genome coverage between 80% and 97%. Samples with a low viral load, as indicated by high Ct values, do not produce enough genome coverage following sequencing due to less viral material in the starting sample. The amPCRe process, which amplifies the target genome using overlapping primers, can help recover genomes of these samples with a low viral load [20]. Through amPCRe, the target genome is amplified to generate sufficient DNA for sequencing, resulting in improved whole genome sequence recovery. As shown through comparison by percent genome coverage, CCE remains superior in terms of genome coverage. CCE multiplies the virus in a growth medium and exponentially increases its viral load to sufficient levels for optimal genome sequence recovery. However, CCE is an expensive and slow process that cannot be relied upon for near-real-time genomic surveillance in a pandemic situation. Furthermore, CCE is unpredictable and has a tendency of occasionally failing to propagate the virus even when samples of reasonably viral titers have been used.

To capture sufficient RVFV material for genome assembly cost-effectively and in a timely manner, we employed an amPCRe approach before sequencing. This technology relies on nucleic acid amplification of the viral genome fragments in single multiplex PCR reactions. The amPCRe method generated enough reads that aligned to the RVFV reference genome. The use of an optimal annealing temperature (T_a_) in a PCR reaction is critical to ensure specificity and efficiency [53]. Determination of the optimal annealing temperature is often dictated by the PCR primers and the melting temperatures (T_m_) of the products. Regions of DNA which have a high melting temperature and are rich in GC content are susceptible to the formation of secondary structures [53]. Consequently, finding the appropriate melting temperature for primer-template annealing is critical. The average GC content of the S segment as indicated by the amplicon primers is slightly lower than that of the reference genomes. Consequently, regions of lower GC content result in optimal amplification of the nucleic acid, thereby yielding sufficient material for sequencing (Figure 3G). Amplicon sequence data are further normalized by trimming off the tiling primers [41], which may affect mapping. The implication of the PCR parameters and the normalization of the sequence data can further explain the observed differences in read depth and coverage between the genome segments.

A hallmark of amplicon sequencing is coverage dropout observed at the ends of the genome segments (5′ and 3′ untranslated regions (UTRs)) compared to Direct or CCE-treated samples. The coverage dropout corresponds to the overlapping nature of amplicons in the multiplex PCR method and is therefore expected. The coverage dropout in the 3′ and 5′ untranslated regions falls outside the outer primer binding sites and is therefore less likely to be amplified at the same rate as other parts of the genome. The 3′ and 5′ ends of sequences can disproportionately affect the successful recombining and extension of DNA strands. As such, while designing primers for such regions, mismatches are severely penalized. Additionally, we observed a consistent coverage dropout in the intergenic region (approximately 90 bp) of the S segment in both enriched and non-enriched samples. The primers S_2_RIGHT (spanning positions 709–739) and S_4_LEFT (spanning positions 957–979), which amplify the intergenic region (positions 845–915) in the S segment, did not generate sufficient amplicons for sequencing, hence the drop in coverage (Figure 3F). These primers amplify the intergenic region separating the NSs and NP genes. This region is regarded as an inaccessible part of the genome, characterized by homopolymeric stretches of Cs that make it difficult to amplify by PCR due to the repetitions of the same nucleotide [23].

Reconstructing viral genomes with high coverage and depth is crucial in phylogenetic inference and other genome-based genetic studies. Molecular clock evolutionary analysis revealed that the virus genomes generated by the three treatment options can be used to infer the genetic evolution of the virus. The amplicon sequencing approach developed in this study generated viral genomes with a mean genome coverage of >80%, sufficient for genotyping purposes. The high concordance in SNPs called in amplicon-enriched and direct or cultured samples shows the suitability of our method in evolutionary analysis including phylogenetics and phylodynamics. Among the SNPs identified with high concordance in the three treatment methods were lineage-defining mutations, as shown in a previous study [31]. The lineage-defining SNPs identified in all the genomes generated by the three treatment options were G851A, C881T, G922A, G2005A and G4015A for the L segment; T715A, A1911G and A1913G in the M segment and T101A, C534T, T684C and A758G in the S segment [31,54].

## 5. Conclusions

Although Primal Scheme has been optimized for non-segmented virus genomes, here, we showed that a multiplex PCR primer scheme can also be generated for segmented viruses. We have shown that the amplicon enrichment method for sequencing RVFV clinical samples with low viral titers, as indicated by Ct values of less than 30, can be reliably used in generating near-complete genomes of the virus. Where CCE fails in samples with moderate Ct values, amPCRe has the advantage of recovering the genomes at levels useful for genotyping. However, obtaining full-length whole genomes in samples with high Ct values (>30) remains a challenge. Importantly, the timing of sampling during an outbreak is key in ensuring adequate viral load for successful sequencing. Therefore, there is a need for continued advancement and execution of best procedures for handling samples without disrupting standard clinical workflows for wider adoption in genomic surveillance during outbreaks. Although amPCRe has proved to be successful in the recovery of genomes from isolates with Ct values < 30, further optimization is needed to reduce the coverage dropouts observed in specific regions of the genome. Genome recovery following amPCRe does not ordinarily generate 100% coverage; however, sufficient sequence coverage is obtained, suitable for genotyping and other genomic epidemiological studies such as transmission dynamics.

## Figures and Tables

**Figure 1 viruses-15-00477-f001:**
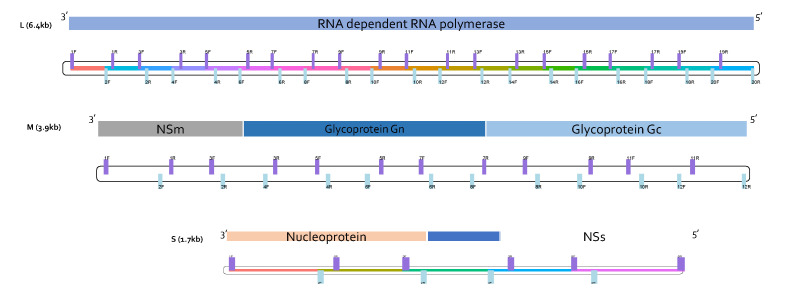
Schematic representation of RVFV primer schemes. The three genomic segments for RVFV and the primers’ mapping positions are shown. Forward primers have the suffix ‘F’ and are colored in purple, while reverse primers are suffixed with ‘R’ and colored in light blue. For each segment, primers were designed with Primal Scheme [20] for the generation of amplicons with a target size of 400 bp in the genome. In total, there were 74 primers (38 for pool 1 and 36 for pool 2 reactions). There were 20, 12 and 6 primer pairs for the L, M and S segments, respectively. The minimum amplicon size for the primers was 355 and the maximum size was 374.

**Figure 2 viruses-15-00477-f002:**
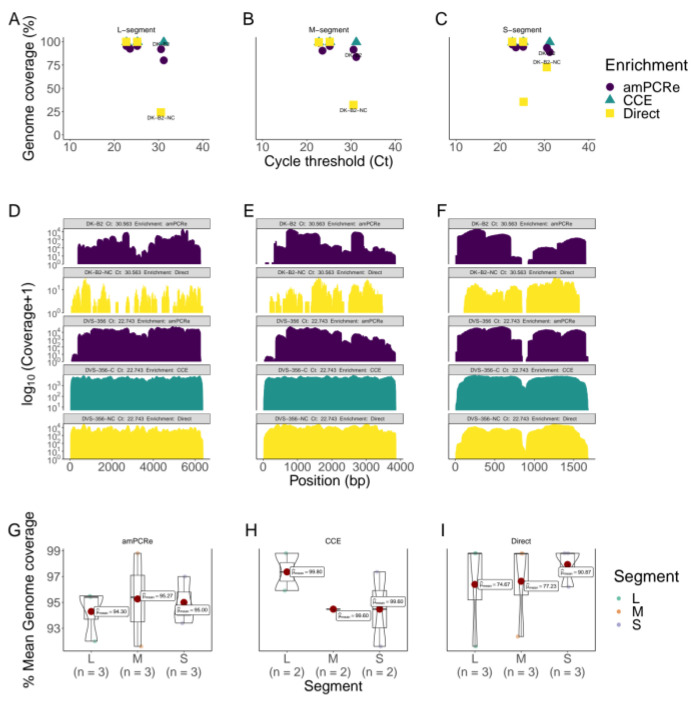
Genome coverage and RT-qPCR. RVFV-positive samples’ cycle threshold (Ct) values (*x*-axis) versus genome coverage (*y*-axis) for the (**A**) large, (**B**) medium and (**C**) small segments. Samples plotted on the graph (DVS-230, DVS-356, DK-B2, RU1 and 08HAB); amPCRe samples are colored in purple, CCE samples are colored in green and Direct samples are shown as yellow. Lower Ct values indicate higher viral titers, while higher Ct values depict low viral titers. CCE, Direct and amPCRe samples are represented using triangle, square and circle shapes, respectively. Genome-wide density plots show coverage (log10-transformed) across the RVFV genome segments: (**D**) a 6404 bp L segment, (**E**) a 3885 bp M segment and (**F**) a 1690bp S segment. The coverage density plots are colored as purple, yellow and green to represent the different treatments. There is a significant drop in coverage in the middle of the S segment, an area characterized by homopolymers of C bases. Box plots and violin plots show the mean genome coverage of the consensus sequences for all the virus segments (L, M and S) in (**G**) amPCRe, (**H**) CCE and (**I**) Direct samples.

**Figure 3 viruses-15-00477-f003:**
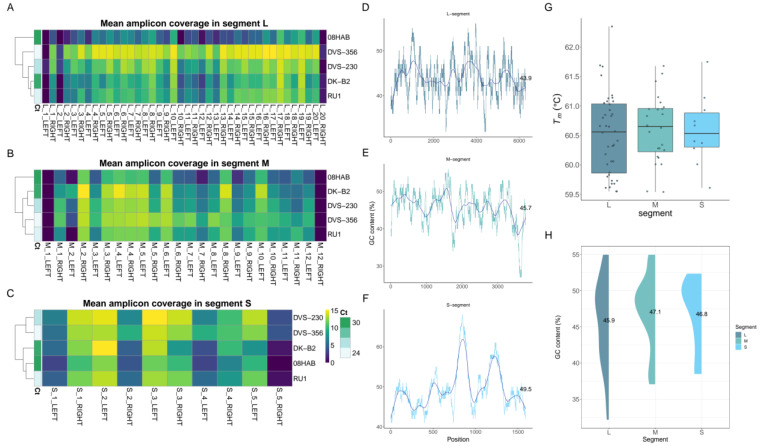
Amplicon primer coverage for RVFV. Primal Scheme generated 74 sets of primers: small (*n* = 10), medium (*n* = 24) and large (*n* = 40). Amplicon primers with odd numbers on their prefix belong to pool 1, while those with even numbers are pool 2 primers. Panel (**A**) shows heatmaps of the mean amplicon coverage across the L segment, (**B**) M segment and (**C**) S segment per sample. The first and last amplicon primers did not sufficiently amplify the 5′ and 3′ ends of the sequences, thereby resulting in a drop in coverage. The dark purple color on the heatmaps show primers which resulted in minimal amplification, while bright yellow color indicates sufficient coverage following amplification with both primer pools. Panels (**D**–**F**) show the percentage GC content computed in a sliding window of 100 bp along the reference genome (ZH-548 strain) for the L, M and S segments, respectively. The solid blue line in the middle depicts the average GC%. Panel (**G**) shows the box plots of the melting temperature, and panel (**H**) indicates the distribution of percent GC content for all the amplicon primers per genomic segment.

**Figure 4 viruses-15-00477-f004:**
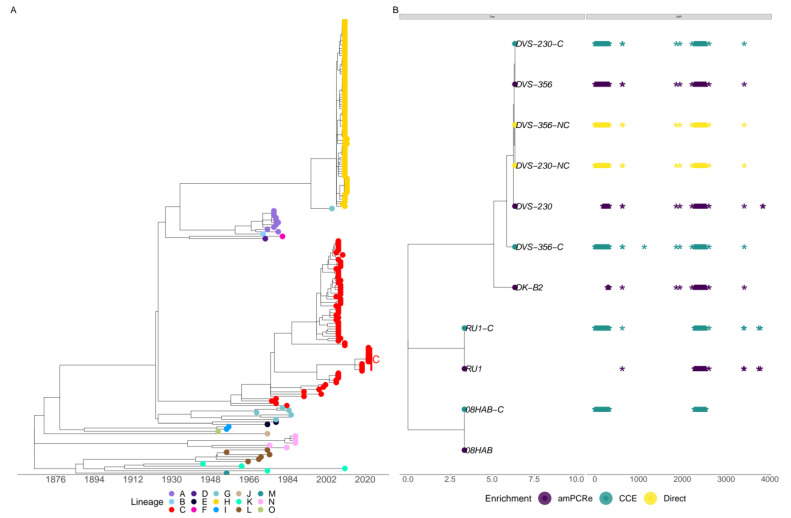
Molecular clock phylogenetic reconstruction. (**A**) Time-scaled phylogenetic tree of heterochronous sequences (*n* = 196) belonging to the M segment, showing the occurrence of lineages from 1951 to 2021. The tips of the phylogenetic tree are colored according to the lineages. The clade labeled C shows the placement of the 11 genomes from this study within the global context. (**B**) Sub-tree of the 11 genomes generated in the study, indicating that all the genomes, irrespective of the treatment option, confidently place the sequences in the global tree, assigned as lineage C.

**Table 1 viruses-15-00477-t001:** Samples used in this study were subjected to three treatments: amplicon multiplex PCR enrichment (amPCRe), cell culture enrichment (CCE) and non-enrichment (Direct). The samples comprised archived as well as clinical outbreak samples from bovine species as hosts for RVFV.

Sample ID	Treatment(s)	Host Species	Sample Type	Country	Location	Collection Date
DVS-230	amPCRe, CCE, Direct	Bovine	Serum	Kenya	Kiambu	2021
DVS-356	amPCRe, CCE, Direct	Bovine	Serum	Kenya	Kiambu	2021
DK-B2	amPCRe, CCE, Direct	Bovine	Serum	Kenya	Murang’a	2021
RU1	amPCRe, CCE, Direct	Bovine	Serum	Rwanda	Rulindo	2018
08HAB	amPCRe, CCE, Direct	Bovine	Serum	Kenya	Wajir	2018

**Table 2 viruses-15-00477-t002:** Primers and probe set used for RT-qPCR assay [32].

RVFV Segment	Primer Name	Sequence 5′–3′
L	RVFL-2912fwdGG	TGAAAATTCCTGAGACACATGG
L	RVFL-2981revAC	ACTTCCTTGCATCATCTGATG
L	RVFL-probe-2950	CAATGTAAGGGGCCTGTGTGGACTTGTG

**Table 3 viruses-15-00477-t003:** Amplicon multiplex PCR components.

Component	Amount (µL)	Final Concentration
Q5 Hot Start Master Mix buffer *	12.5	1×
Primer pool 1 or 2 (10 µM)	1.4 each for pool 1 and pool 2	0.015 µM per primer
Nuclease-Free Water	Up to 7 µL	
cDNA	4.5	

## Data Availability

All sequence data generated from genome reconstruction and used in the study are available on OM744365–OM744379. All the sequence accessions are provided in the Appendix A.

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
