# Peer review of "Using Multiplex Amplicon PCR Technology to Efficiently and Timely Generate Rift Valley Fever Virus Sequence Data for Genomic Surveillance"

_viruses, 2023, doi:10.3390/v15020477_

Round 1

Reviewer 1 Report

In my opinion the work is very interesting and very clearly exposed, so for sure it will be quite useful for those researchers interested in RVFV. My suggestions to improve the paper are as follows:

(i) Regarding Methods section: I would appreciate a more detailed - explicit description on the method for sequencing followed for those samples that are not amPCR enriched : cDNA, library preparation are done with the same kits? 

(ii) Regarding results: although maybe non critical for sequencing focused on isolates genotyping, SNP results are very important for other studies, as authors remarked, so I find it very interesting to know about possible changes associated to the different methods used (cell culture growing, direct analyses or amPCRe). For a better comparison and understanding, I would appreciate to see those results described in section 3.3 in a table (as supplementary information would be enough)

Reviewer 2 Report

The paper submitted described a multiplex amplicon approach for sequencing Rift Valley fever virus. However, certain weaknesses in the experiment design had an impact on the final result.

Major:

Because LunaScript uses an oligo-dT and random hexamer method for cDNA synthesis, it leads to limited coverage of the 5' end and 3' end, as seen in Figure 2. Custom RT primers would alleviate this issue.

For illumina short read sequencing, the number of clusters and reading length are not described in method. It is uncertain if the authors normalized sequencing clusters for a fair comparison. 

It would be great to include a schematic graph of the viral genome and primer pair location.

Minor:

Some of the annotations in the figures are too small to read.

Treatments in Table 1 make use of both CCE and culture. Please use consistent terminology when discussing the same approach.

Reviewer 3 Report

Great manuscript! Just a few minor comments and suggestions below.

Introduction:

page 2 line 73: add "virus" after Zika

Materials and methods:

page 3 table 3: use CCE thoughout and not "Culture"

page 4 line 145: define "BSL3"

page 4 line 146: "media" instead of "Media"

page 4 line 157: what does the authors mean by "harvested"?

page 4 line 172-173: please review this sentence - how much template was used?

page 4 line 173-175: please review this sentence - suggest to move "40 cycles" from the middle of the sentence, since it does not include the annealing/elongation step

page 5 line 182-183: the sentence ".[25]... for the primers" needs to be reviewed

page 5 line 194: was the cDNA step done as per manufacturer's instructions?

page 5 line 202: define "TE buffer"

page 5 line 198-207: suggest to revise this section, not very clear how the amplicon multiplex PCR was set up - how many primers were included in each pool - where these seperate for each segment?

page 5 table 3 (and throughout the document): be consistent with decimal places, choose a format and stick to it (ie 12.5 vs 1.425 vs 1.35)

page 5 line 213: add company, city, country for NEBNext lib prep kit

page 6 line 233: replace "machine" with "instrument"

page 6 line 246+255: add reference for "viclara" and "MAFFT"

Results:

page 7 line 295: why were sample DK-B2 excluded?

all figures: in the document it is not easy to read the figures, suggest to make them full size in final manuscript

Would be great if the authors can add a schematic representation of the primers on the genome indicating where they are binding

page 9 line 348+352 (and throughout the document): replace RVF virus with "RVFV"

Round 2

Reviewer 2 Report

Thanks for the update.